# Incidences of Deep Molecular Responses and Treatment-Free Remission in de Novo CP-CML Patients

**DOI:** 10.3390/cancers12092521

**Published:** 2020-09-04

**Authors:** Gabriel Etienne, Stéphanie Dulucq, Fréderic Bauduer, Didier Adiko, François Lifermann, Corinne Dagada, Caroline Lenoir, Anna Schmitt, Emilie Klein, Samia Madene, Marie-Pierre Fort, Fontanet Bijou, Marius Moldovan, Beatrice Turcq, Fanny Robbesyn, Françoise Durrieu, Laura Versmée, Sandrine Katsahian, Carole Faberes, Axelle Lascaux, François-Xavier Mahon

**Affiliations:** 1Service d’Hématologie, Institut Bergonié, 33076 Bordeaux, France; A.Schmitt@bordeaux.unicancer.fr (A.S.); M.Fort@bordeaux.unicancer.fr (M.-P.F.); f.bijou@bordeaux.unicancer.fr (F.B.); F.Durrieu@bordeaux.unicancer.fr (F.D.); lversmee@mgh.harvard.edu (L.V.); c.faberes@bordeaux.unicancer.fr (C.F.); Francois-Xavier.Mahon@u-bordeaux.fr (F.-X.M.); 2Institut National de la Santé et de la Recherche Médicale, U1218 ACTION, Université de Bordeaux, 33000 Bordeaux, France; stephanie.dulucq@chu-bordeaux.fr (S.D.); emilie.klein@chu-bordeaux.fr (E.K.); Beatrice.Turcq@u-bordeaux.fr (B.T.); 3Groupe France Intergroupe des Leucémies Myéloïdes Chroniques, Hôpital Haut-Lévêque, 33600 Pessac, France; fbauduer@ch-cotebasque.fr; 4Laboratoire d’Hématologie, Hôpital Haut Lévêque Centre Hospitalier Universitaire de Bordeaux, 33600 Pessac, France; fanny.robbesyn@chu-bordeaux.fr; 5Service d’Hématologie, Centre Hospitalier Côte Basque, 64100 Bayonne, France; 6Collège des Sciences de la Santé, Université de Bordeaux, 33000 Bordeaux, France; 7Service d’Hématologie, Centre Hospitalier de Libourne, 33500 Libourne, France; Didier.adiko@ch-libourne.fr; 8Service de Médecine Interne, Centre Hospitalier de Dax-Côte d’Argent, 40107 Dax, France; lifermann@ch-dax.fr; 9Service d’Oncologie-Hématologie, Centre Hospitalier de Pau, 64000 Pau, France; corinne.dagada@ch-pau.fr; 10Service d’Hémato-Oncologie Radiothérapie, Polyclinique Bordeaux Nord Aquitaine, 33000 Bordeaux, France; C.Lenoir@bordeauxnord.com; 11Service de Médecine Interne et Hématologie, Centre Hospitalier Intercommunal Mont-de-Marsan—Pays des Sources, 40024 Mont de Marsan, France; samia.madene@ch-mdm.fr; 12Service d’Hématologie-Oncologie, Centre Hospitalier de Périgueux, 24000 Périgueux, France; moldovan_marius@yahoo.com; 13Centre National de la Recherche Scientifique, SNC 5010, 33000 Bordeaux, France; 14Massachusetts General Hospital, Boston, MA 02114, USA; 15Unité de Recherche Clinique et Centre Investigation Clinique-Epidémiologie, Hôpitaux Universitaires Paris-Ouest Hôpital Européen Georges Pompidou, Assistance Publique-Hôpitaux de Paris, Université Paris 5 Institut National de la Santé et de la Recherche Médicale, Centre de Recherche des Cordeliers, Equipe 22, 75006 Paris, France; sandrine.katsahian@aphp.fr; 16Service des maladies du sang, Hôpital Haut Lévêque Centre Hospitalier Universitaire de Bordeaux, 33600 Pessac, France; axelle.lascaux@chu-bordeaux.fr

**Keywords:** chronic myeloid leukemia, tyrosine kinase inhibitor, deep molecular responses, treatment-free remission

## Abstract

**Simple Summary:**

Tyrosine kinase inhibitors (TKI) can be safely discontinued in chronic myeloid leukemia patients. Achieving a sustained deep molecular response (DMR) before stop is recommended. Currently, the proportion of patients who achieve a sustained DMR remains to be determined. Based on the follow-up of 398 patients over a ten-years period, we evaluate that 46% of them have achieved a sustained DMR. Gender, BCR-ABL1 transcript type, and disease risk scores were significantly associated with the probability of achieving a DMR. 95/398 (24%) patients stopped TKI with a probability of maintaining molecular reponse without TKI resumption of 47% at 48 months after stop. In this study, TKI duration before stop and second (nilotinib, dasatinib, bosutinib) generation frontline TKI compared to imatinib were significantly associated with a lower risk of molecular relapse after stop in patients who have achieved a sustained DMR.

**Abstract:**

*Background:* Tyrosine Kinase Inhibitors (TKIs) discontinuation in patients who had achieved a deep molecular response (DMR) offer now the opportunity of prolonged treatment-free remission (TFR). *Patients and Methods:* Aims of this study were to evaluate the proportion of de novo chronic-phase chronic myeloid leukemia (CP-CML) patients who achieved a sustained DMR and to identify predictive factors of DMR and molecular recurrence-free survival (MRFS) after TKI discontinuation. *Results:* Over a period of 10 years, 398 CP-CML patients treated with first-line TKIs were included. Median age at diagnosis was 61 years, 291 (73%) and 107 (27%) patients were treated with frontline imatinib (IMA) or second- or third-generation TKIs (2–3G TKI), respectively. With a median follow-up of seven years (range, 0.6 to 13.8 years), 182 (46%) patients achieved a sustained DMR at least 24 months. Gender, *BCR-ABL1* transcript type, and Sokal and ELTS risk scores were significantly associated with a higher probability of sustained DMR while TKI first-line (IMA vs. 2–3G TKI) was not. We estimate that 28% of CML-CP would have been an optimal candidate for TKI discontinuation according to recent recommendations. Finally, 95 (24%) patients have entered in a TFR program. MRFS rates at 12 and 48 months were 55.1% (95% CI, 44.3% to 65.9%) and 46.9% (95% CI, 34.9% to 58.9%), respectively. In multivariate analyses, first-line 2–3G TKIs compared to IMA and TKI duration were the most significant factors of MRFS. *Conclusions:* Our results suggest that frontline TKIs have a significant impact on TFR in patients who fulfill the selection criteria for TKI discontinuation.

## 1. Introduction

Targeting the deregulated tyrosine kinase activity of the BCR-ABL1 fusion protein with selective agents, namely the tyrosine kinase inhibitors (TKIs) in chronic myeloid leukemia (CML) patients constitutes one of the most impressive therapeutic progress over the past 20 years. With the exception of a minority of patients diagnosed in the late phase of the disease or patients who progress on TKIs, most patients achieve a response associated with a very low risk of disease progression and an overall survival close to the healthy population [1,2,3,4]. The main treatment goals in chronic phase CML (CP-CML) patients are now to minimize the risks associated with TKI in the long-term issue and to avoid lifelong TKI exposure.

Several published studies have demonstrated that TKIs can be safely discontinued in CP-CML patients who have achieved a sustained deep molecular response (DMR) from therapy with an overall likelihood of maintaining a major molecular response (MMR) of 40 to 70% after TKI cessation (Review in Dulucq et al. and Cortes et al. [5,6]). Indeed, TKI resumption after the loss of MMR leads to a second DMR in most if not all patients provided that TKI had been restarted and the follow-up after resumption is sufficient [7,8,9,10,11]. The depth and duration of DMR, as well as TKI duration before discontinuation, are essential factors for a successful attempt. Currently, achieving an MR4.5 log reduction in the *BCR-ABL1/ABL1* ratio during at least two years or an MR4 log reduction lasting more than three years before stopping together with an M *BCR-ABL1* transcript, chronic phase at diagnosis and the absence of suboptimal response or failure during TKI treatments are recommended for TKI discontinuation outside a clinical trial [12,13,14]. Less stringent criteria are still under evaluation in some ongoing clinical trials [15,16,17]. It has been estimated that 28 to 36% of CP-CML patients receiving imatinib frontline or after interferon achieved a sustained DMR defined as *BCR-ABL1/ABL1* 4 log (MR4) or 4.5 (MR4.5) log reduction sustained for at least two years [18,19]. Based on the results of frontline randomized clinical trials, second-generation TKIs, i.e., dasatinib and nilotinib ten years ago and more recently bosutinib, have been approved as first-line therapeutic options [20,21,22]. Final updates of the Dasatinib vs. the Imatinib Study in Treatment-Naïve Chronic Myeloid Leukemia Patients Trial (DASISION) and the Evaluating Nilotinib Efficacy and Safety in Clinical Trials-Newly Diagnosed Patients (ENESTnd) demonstrated that frontline second-generation TKIs were associated with higher cumulative incidences of DMR compared to imatinib [23,24].

However, maintenance and duration of these DMR are unknown. Hence, currently, the proportion of CML patients receiving frontline TKIs who achieve a sustained DMR needs to be re-evaluated. We conducted a multicenter, retrospective, and observational study. The primary objective was to assess in de novo TKI CP-CML patients the incidence and predictive factors of sustained DMR. The secondary objective was to evaluate the rate and predictive factors of molecular recurrence-free survival (MRFS) in patients who discontinued TKIs.

## 2. Methods

### 2.1. Patients

From January 2006 to December 2015, we included adult patients (≥18 years old), diagnosed in CP of the disease and treated with any first-line TKI at any dose in the 9 southwest French participating centers. Patients who received frontline TKIs within clinical trials were not excluded. All patients agreed to be included in this observational study and provided informed consent in accordance with the Declaration of Helsinki.

### 2.2. Definitions

Chronic, accelerated, and blastic phases of the disease (CP, AP, and BP) as well as responses, overall and progression survivals were defined in line with the ELN recommendations [25,26]. Deaths were considered CML-related when occurred in AP or BP and related to the progression of the disease. Sokal and EUTOS Long-Term Survival (ELTS) risk scores were calculated according to the proposed formulations [27,28]. The molecular response was assessed by reverse transcription-quantitative polymerase chain reaction (RT-qPCR) and reported as ratios of *BCR-ABL1* to *ABL1* standardized to the international scale (IS) (*BCR-ABL^IS^*) according to previously reported recommendations [29,30]. All *BCR-ABL1/ABL1* assessments were performed in a unique central laboratory belonging to the Groupe des Biologistes Moléculaires des Hémopathies Malignes (GBMHM) and European Treatment Outcome Study (EUTOS) labs group. *BCR-ABL1/ABL1* assessments were performed every 3 to 4 months and MMR, MR4, and MR4.5 were defined as 3-log reduction or *BCR-ABL1^IS^* ≤ 0.1%, 4-log reduction or *BCR-ABL1^IS^* ≤ 0.01%, and 4.5-log reduction or *BCR-ABL1^IS^* ≤ 0.0032%, respectively.

Each molecular response had to be confirmed on two consecutive analyses at least two months apart. The date of each molecular response corresponds to the first date of obtaining a molecular response. MR4.5 was defined as sustained if no more than one *BCR-ABL1^IS^* assessment in a twelve-month period demonstrated an MR4.5 loss with retained MR4. Molecular follow-up after TKI discontinuation was performed as recommended [13]. Molecular recurrence after TKI discontinuation was defined either as a confirmed positivity of *BCR-ABL1^IS^* with an increase of one log on two consecutive assessments or MMR loss. In the case of molecular recurrence, restart TKI used before discontinuation was recommended. Molecular recurrence-free survival was measured from the date of TKI stop to the date of the first event, i.e., molecular recurrence or death, or to the date of last available follow-up.

### 2.3. Statistical Analyses

Quantitative variables were described using the median and the range (min – max). For qualitative parameters, the results were presented as numbers (proportions). Qualitative and quantitative variables were compared according to frontline TKI type or the occurrence of molecular recurrence using Chi-square or Fisher exact tests for qualitative variables and Student’s *t*-test or Wilcoxon rank-sum test for quantitative parameters. Baseline patients’ characteristics such as gender (female vs. male), age, *BCR-ABL1* transcript (B3A2 vs. each type), additional cytogenetic abnormalities vs. none, Sokal and ELTS (low vs. Intermediate, low vs. high) risk scores, first-line TKI (IMA vs. 2–3G TKI) were analyzed as potential factors for achieving MR4.5, MR4.5 sustained more than 24 months, or for experiencing molecular recurrence after TKI discontinuation. Ongoing characteristics (e.g., duration of TKI treatment, time to MR4.5, MR4.5 duration, history of TKI switch due to intolerance or lack of efficacy) were also tested as prognostic factors for molecular recurrence. Associations between MR4.5 and sustained MR4.5 and the variables were analyzed using the competing risks method. Allogenic Hematopoietic Stem Cell Transplantation (AHSCT) or death without DMR were considered as competing events. The effect of covariates was estimated using the Fine and Gray model. The proportional hazard assumption was tested using the Lin test. Univariate and multivariate analyses were performed using Cox regression to analyze factors associated with the molecular recurrence because of no competing events to the occurrence of molecular recurrence. Variables were included in the multivariate analysis if they were significantly associated with the outcome in the univariate analysis (*p* < 0.05) or if they were clinically relevant based on previously published results.

To establish factors associated with molecular recurrence, patients with TKI cessation for efficacy only and with a follow up of at least 6 months after discontinuation were analyzed.

Results were presented as subdistribution hazard ratios (SHR) with their 95% confidence interval.

Cumulative incidences of DMR (MR4.5 and sustained MR4.5) were measured from frontline TKI start to the date of confirmed MR4.5 or to the date from which MR4.5 remained sustained and were compared using the Gray test. Molecular recurrence-free survival (MRFS) was measured from the date of TKI discontinuation to the date of molecular recurrence or the last follow-up for patients who did not relapse. MRFS was calculated using the Kaplan–Meier method and survival differences were assessed using a log-rank test. Progression-free survival (PFS) from TKI start to the day of AP or BP phase or death, and overall survival (OS) were estimated using the Kaplan–Meier method.

All statistical analyses were carried with R software and IBM^®^ SPSS software, version 22 (IBM Corp., Armonk, NY, USA). The statistical significance level was (two-sided) 0.05.

## 3. Results

### 3.1. Patients

From January 2006 to December 2015, 398 CML patients in CP at diagnosis were recruited in the nine southwest French participating centers and were included in the study. The baseline characteristics of the study population are presented in Table 1. The median age at diagnosis and the proportion of high Sokal risk score significantly differ between patients receiving frontline 2–3G TKIs or IMA.

### 3.2. Outcome

The median follow-up from diagnosis to last date of follow-up or death was seven years (range: 0.6 to 13.8). Seventeen (4.2%) patients have progressed to AP or BP of the disease. Sixteen (4%) patients underwent AHSCT while in the first chronic phase (*n* = 8) or after progression to AP or BP (*n* = 8), whereas one patient progressed to myeloid blast crisis during year six—all the progression events occurred during the four years following CML diagnosis. Sixty (15%) patients died. Ten deaths were considered as CML-related (myeloid blast crisis in nine patients, lymphoid in one). Fifty patients died due to CML-unrelated causes (secondary neoplasm, *n* = 18; cardiovascular diseases, *n* = 10; non-cardiac chronic organ failures, *n* = 13; death related to AHSCT, *n* = 4; infectious diseases, *n* = 3; suicide, *n* = 1; unknown, *n* = 1). Five- and 10-year overall and progression-free survivals are 90% (95% confidence interval (CI 95%) CI, 87.06% to 92.94%) and 81% (95% CI, 76.0% to 85.8%), and 88% (95% CI, 85.16% to 91.44%) and 79% (95% CI, 74.4% to 84.2%), respectively.

#### 3.2.1. Cumulative Incidences of Deep Molecular Responses (MR4.5 and Sustained MR4.5 at Least 24 Months)

Overall, 349 (87%) patients achieved an MMR, 290 (72%) an MR4, and 250 (62%) an MR4.5. The median times from TKI start to molecular response were 9.3 months (range: 1.4 to 84.2) for MMR, 23.5 months (range: 2.6 to 115.3) for MR4, and 30.1 months (range: 2.6 to 145.5) for MR4.5. The proportion of patients still on first-line TKI was 80% at the time of MMR, 75% at the time of MR4, and 75% at the time of MR4.5. Incidences of MMR, MR4, MR4.5, and sustained MR4.5 of at least 24 months and TKI discontinuation are summarized in the flow diagram in Figure 1.

Finally, 182 (46%) patients experienced a sustained MR4 with a median time from TKI start to the onset of sustained MR4.5 of 25.5 months (range: 2.6 to 106.7). The cumulative incidence of sustained MR4.5 for the whole study population is presented in Figure 2A.

#### 3.2.2. Predictive Factors of Deep Molecular Responses (MR4.5 and MR4.5 Sustained for at Least 24 Months)

Predictive factors of MR4.5: By univariate analysis, female gender (*p* = 0.0097), B3A2 vs. B2A2 (*p* = 0.0056), low vs. intermediate Sokal (*p* = 0.01), low vs. high Sokal (*p* < 0.001) risk, low vs. intermediate ELTS (*p* < 0.001), low vs. high ELTS (*p* < 0.001) risk, were statistically significantly associated with a higher MR4.5 cumulative incidence whereas only a trend was observed for patients without ACA vs. others (*p* = 0.065). Frontline TKI type was not statistically significant (*p* = 0.22).

In the final multivariate analysis model, the following were associated with a higher cumulative incidence rate: female gender (SHR ± 95% CI: 1.44 ± 0.27, *p* = 0.008), B3A2 vs. B2A2 (SHR ± 95% CI: 1.50 ± 0.29, *p* = 0.006), low vs. intermediate ELTS risk (SHR ± 95% CI: 2.28 ± 0.32, *p* < 0.001), low vs. high ELTS risk (SHR ± 95% CI: 2.91 ± 0.48, *p* < 0.001). We observed similar results with the Sokal risk score in the final multivariate analysis model.

Predictive factors of sustained MR4.5 for at least 24 months: The same variables as for MR4.5 were analyzed to identify predictive factors of sustained MR4.5 of at least 24 months. Univariate analysis showed that the female gender (*p* = 0.021), low vs. intermediate Sokal (*p* = 0.016), low vs. high Sokal (*p* < 0.001) risk, low vs. intermediate ELTS (*p* < 0.001), and low vs. high ELTS (*p* = 0.003) risk, were statistically associated with a higher sustained MR4.5 cumulative incidence (Table 2). We observed only a trend for B3A2 vs. B2A2 *BCR-ABL1* transcript type (*p* = 0.082). The cumulative incidence of sustained MR4.5 for at least 24 months according to TKI first-line (IMA vs. 2–3G TKI) is presented in Figure 2B. No statistically significant difference between the two TKI treatment groups was demonstrated (*p* = 0.96). At the onset of sustained MR4.5 for at least 24 months, the majority of patients were still receiving the first-line TKIs (76.6% and 85.1% for IMA and 2–3G TKI, respectively). Only 6.6% of patients treated with IMA frontline had switched to 2–3G TKI for failure or suboptimal response according to the 2013 ELN definitions [26]. In multivariate analyses, the following factors retained statistical significance: female gender (SHR ± 95% CI: 1.40 ± 0.30, *p* = 0.028), low vs. intermediate ELTS risk (SHR ± 95% CI: 2.18 ± 0.37, *p* < 0.001), low vs. high ELTS risk (SHR ± 95% CI: 2.10 ± 0.50, *p* = 0.004). In the multivariate model where ELTS was replaced by Sokal risk score, similar results were observed. Finally, we identified three potential baseline risk factors for the occurrence of sustained MR4.5 for at least 24 months: female gender, B3A2 *BCR-ABL1* transcript type, and low ELTS (or Sokal) risk score. The cumulative incidence of sustained MR4.5 for at least 24 months reached 85% in female patients with a B3A2 *BCR-ABL1* transcript type and a low ELTS risk score. In contrast, the cumulative incidence observed in male patients harboring a non B3A2 *BCR-ABL1* transcript type and an intermediate or high ELTS risk score was only 30%. The cumulative incidence of sustained MR4.5 of at least 24 months in patients presenting with either none, one, two, or three of the latter potential risk factors are presented in Figure 2C.

#### 3.2.3. Proportion of Patients Who Stopped TKI and Molecular Relapse-Free Survival

Among the 182 patients who achieved a sustained MR4.5 at least 24 months, 84 (46%) patients have TKI discontinuation for efficacy with a TFR objective. Eleven additional patients discontinued TKIs for efficacy, six after obtaining an MR4.5 sustained less than 24 months and five without a prior sustained MR4.5. Most patients (75%) who stopped TKI were followed in a university hospital.

Thus, 95/398 (24%) patients have TKI discontinuation for efficacy. Among these 95 patients, 83 patients were included in the TFR analyses. Twelve patients, with a follow-up lower than 6 months after TKI discontinuation were excluded from the TFR analyses. The baseline and ongoing characteristics of the 83 patients are presented in Table 3. Among them, 42/83 patients had molecular relapse consisting of MMR loss in 36 patients and *BCR-ABL1^IS^* increase of at least 1 log with sustained MMR in six patients. MRFS rates at 12 and 48 months were 55.1% (95% CI, 44.3% to 65.9%) and 46.9% (95% CI, 34.9 to 58.9%), respectively (Figure 3A).

Among the 83 patients included in the TFR analyses, 61 (73.5%) patients received first-line IMA, and 22 (26.5%) first-line 2–3G TKIs. At the time of TKI discontinuation 15 patients treated with first-line IMA had switched to 2–3G TKIs (10 for intolerance to IMA and 5 because of warning or failure) with a median time on IMA of 7.4 months (range, 1.4 to 99.3 months) and 3/22 patients treated with first-line 2–3G TKI had switched to another 2–3G TKI (intolerance to first-line 2–3G TKI in 3). Among the patients who had a previous switch, eight patients had *BCR-ABL1* tyrosine kinase mutation analyses (using Next Generation Sequencing). Among them, five patients had a previous history of warning or failure and three patients had TKI switch due to intolerance without MMR at the time of switch. None of these patients harbored a *BCR-ABL1* domain tyrosine kinase mutation. The median follow-ups from the date of TKI discontinuation to the date of last visit for patients with or without molecular relapse were 26.0 months (range: 11.9 to 78.7) and 34.6 months (range: 7.3 to 99.2), respectively. Three patients died of CML-unrelated causes (cirrhosis, *n* = 1; myocardial infarction, *n* = 1; infectious disease, *n* = 1). All patients died after molecular relapse and TKI resumption.

By univariate analysis, among the patients’ baseline and ongoing characteristics analyzed to identify predictive factors of MRFS, first-line TKI (2–3G TKI vs. IMA), was statistically significantly associated with a better MRFS and a trend was observed for TKI duration and MR4.5 duration.

We performed 2 different multivariate analyses adjusted either on TKI duration or on MR4.5 duration. Considering that some patients receiving first-line IMA had switched to 2–3G TKI in case of intolerance or lack of efficacy, we included in multivariate analysis the history of a switch before TKI discontinuation (Table 4, additional Table 5). The duration of TKI treatment and frontline 2–3G TKIs were the most significant factors. Patients with frontline 2–3G TKI whether or not they had switched, had near three times less risk of molecular relapse than those treated with IMA frontline (HR: 0.36; 95%CI: 0.15 to 0.85; *p* = 0.027) (Table 4, Figure 3B). MRFS according to frontline TKIs and previous switch to other TKIs are presented in Figure 3C. If we take into account patients with MMR loss only as a definition of molecular relapse, the same results were observed (data not shown).

## 4. Discussion

With a median follow-up of seven years, the 10-year estimated OS and PFS observed in the study cohort were 81% and 79%, respectively. These results are similar to the findings reported in previous clinical trials evaluating IMA alone or IMA-based regimens in de novo CP-CML patients [3,4]. As expected and due to the age of our study population, the rate of CML-unrelated death (83% in this study) is higher than that reported in younger CML patient cohorts. Conversely, the incidences of progression events and AHSCT in the first chronic phase—which may prevent progression to late phases of the disease in several patients—were less frequent in our patient’s population than others despite a similar proportion of high Sokal and ELTS risk scores. These differences may be explained in part by the recruitment period (2006 to 2015) with the access to 2nd and 3rd generation TKIs (as frontline treatment or in the case of intolerance or resistance to IMA). Overall, 250 (62.8%) patients had obtained an MR4.5. Among them, 188 (75%) patients were still on first-line TKIs whereas 25% had switched to another TKI before achieving MR4.5 because of TKI intolerance rather than an unsatisfactory response. In the group of the 68 patients who did not achieve sustained MR4.5 for at least 24 months, approximately half of them had a follow-up between the onset of MR4.5 and the last *BCR-ABL1*/*ABL1* available assessment below 24 months. These patients will probably achieve a further stable MR4.5 as the median delay to obtain sustained MR4.5 is 30 months in our cohort.

Provided that only 27% of the patients included in this study were treated with 2–3G TKIs, the cumulative incidence of MR4.5 is comparable to the rates in patients with available molecular assessment from the IRIS trial and the CML-Study IV [3,31]. As sustained MR4.5 were recorded whatever the TKI at the time of the onset of MR4.5 together with a longer follow-up may explain the higher cumulative incidence of stable MR4.5 in this study compared to previous evaluation reported by Branford et al. seven years ago [18].

Several characteristics were associated with a higher probability of achieving an MR4.5 and sustained MR4.5 of at least 24 months. As previously reported, the female gender, B3A2 *BCR-ABL1* transcript type, and low-risk scores were associated with a higher probability of MR4.5 [18,32,33,34]. Surprisingly, first-line 2–3G TKI (vs. IMA) was not associated with a higher probability of achieving a sustained MR4.5. Several explanations may be suggested. On the one hand, 28% of the 291 patients treated with first-line IMA had switched to 2–3G TKI before achieving MR4.5 leading to a higher rate of DMR as previously reported [35,36]. On the other hand, patients receiving first-line 2–3G TKI were significantly younger than those receiving first-line IMA treatment (53 years vs. 63 years, *p* < 0.001). Taking into account that CML phenotype at diagnosis is more aggressive in children and young adults than in older patients [37,38], and that CML genotype including *BCR-ABL1* transcript type and somatic variants may also differ with a higher incidence of B2A2 transcript and some (un)known genetic variants in young patients [34,39,40,41]. These differences may be accountable for higher rates of IMA failures and lower rates of DMR observed in the young adult populations than the older population [32,42]. This negative impact of young age on the disease course seems to be reduced with first-line 2nd generation TKIs [42,43,44]. Finally, we believe that the absence of a significant difference in terms of cumulative incidence of DMR between IMA and 2–3G TKI first-line observed in this study is related to the fact that younger patients were preferentially treated with 2–3G TKIs and older patients with first-line IMA.

One of the objectives of this study was to evaluate the proportion of CP-CML patients treated with frontline TKIs eligible for TFR. Among the 398 patients included in this study, 250 (62.8%) patients have achieved an MR4.5, and 182 (46%) a sustained MR4.5 of at least 24 months. Among these 182 patients, 72 patients had a TKI duration below five years and or a prior history of failure or warning according to the 2013 ELN recommendations [26]. Finally, 110/398 (28%) patients fulfilled the criteria for TFR proposed by the French Chronic Myeloid Leukaemia Study Group (Fi-LMC) [13]. Approximately half of the patients who achieved a sustained MR4.5 had entered in a TFR phase. Most patients included in a TFR phase were treated in a university hospital illustrating that TFR evaluation was first performed inside clinical discontinuation trials available only in university hospitals and started later in non-university centers as soon as TFR study results and or recommendations were available.

MRFS at 12 and 48 months after TKI discontinuation were estimated at 55.1% (95% CI, 44.3% to 65.9%) and 46.9% (95% CI, 34.9 to 58.9%), respectively. As no patient resumed TKI without molecular relapse and no death was observed in patients without molecular relapse, the probabilities of TFR and MRFS do not differ.

Although both studies are not comparable, MRFS observed in our study appeared to be lower than the one reported in the previous year by Fava et al. [45]. We hypothesize that baseline characteristics of the populations with a higher proportion of low-risk Sokal score (59% vs. 24%) and a longer TKI duration before stop (7.2 vs. 5.7 years) may account for these differences.

Interestingly, in multivariate analysis first-line TKIs (IMA vs. 2–3G TKIs) in addition to TKI duration were the most predictive factors of MRFS regardless of the definition of molecular recurrence leading to TKI resumption. Whether *BCR-ABL1* dependent or independent mechanisms reported to be associated with leukemic stem cells’ persistence leading to molecular relapse after TKI discontinuation are differently impacted by TKIs remains to be determined. However, our results suggest that frontline 2–3G TKIs compared to IMA induce more frequent durable DMR after TKI discontinuation.

In this study, we have analyzed the cumulative incidence of DMR and MRFS after TKI stop taking into account the frontline TKI type and compared IMA vs. other available TKIs inside or outside clinical trials. This comparison remains debatable. However, distinct randomized frontline clinical trials have compared 2nd (nilotinib, dasatinib, bosutinib) and 3rd (ponatinib) generation TKIs to IMA [22,23,24,46]. Except for ponatinib and to a lesser extent bosutinib, 2nd generation TKIs lead to significant sustained higher rates of MMR, MR4, and MR4.5 when compared to IMA [23,24]. Currently, as no head to head comparison between 2nd generation TKI has been done in the frontline setting, we hypothesized that the incidences of DMR would not significantly differ between 2nd and 3rd generation TKIs. In addition, discontinuation of 2nd generation TKIs as front-line or after IMA lead to similar rates of molecular recurrence-free remission [11,47,48,49,50,51].

## 5. Conclusions

In conclusion, whatever the type of frontline TKI, nearly 30% of CP-CML patients receiving first-line TKIs would be optimal candidates for TFR. Due to the different choices of first-line TKI mainly based on a patient’s age at diagnosis, first-line 2–3G TKIs and IMA lead to the same rates of DMR. However, when patients entered into a TFR phase, our results suggested that TKI duration and treatment by first-line 2–3G TKIs were both associated with higher MRFS.

## Figures and Tables

**Figure 1 cancers-12-02521-f001:**
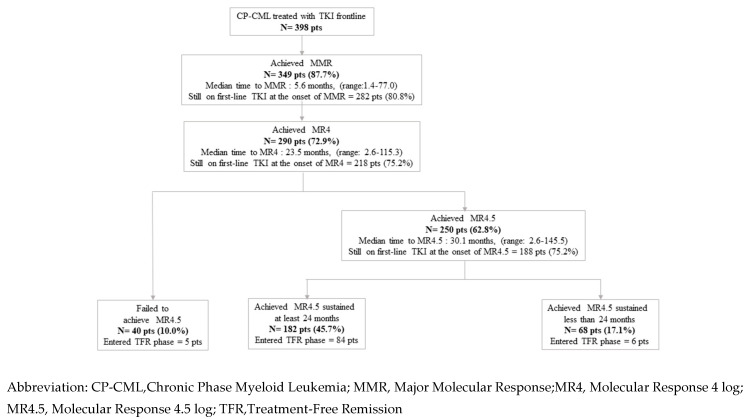
Flow diagram of molecular response (major molecular response, molecular response 4 log, molecular response 4.5 log, molecular response 4.5 log sustained at least 24 months) and TKI discontinuation (N = 398 pts).

**Figure 2 cancers-12-02521-f002:**
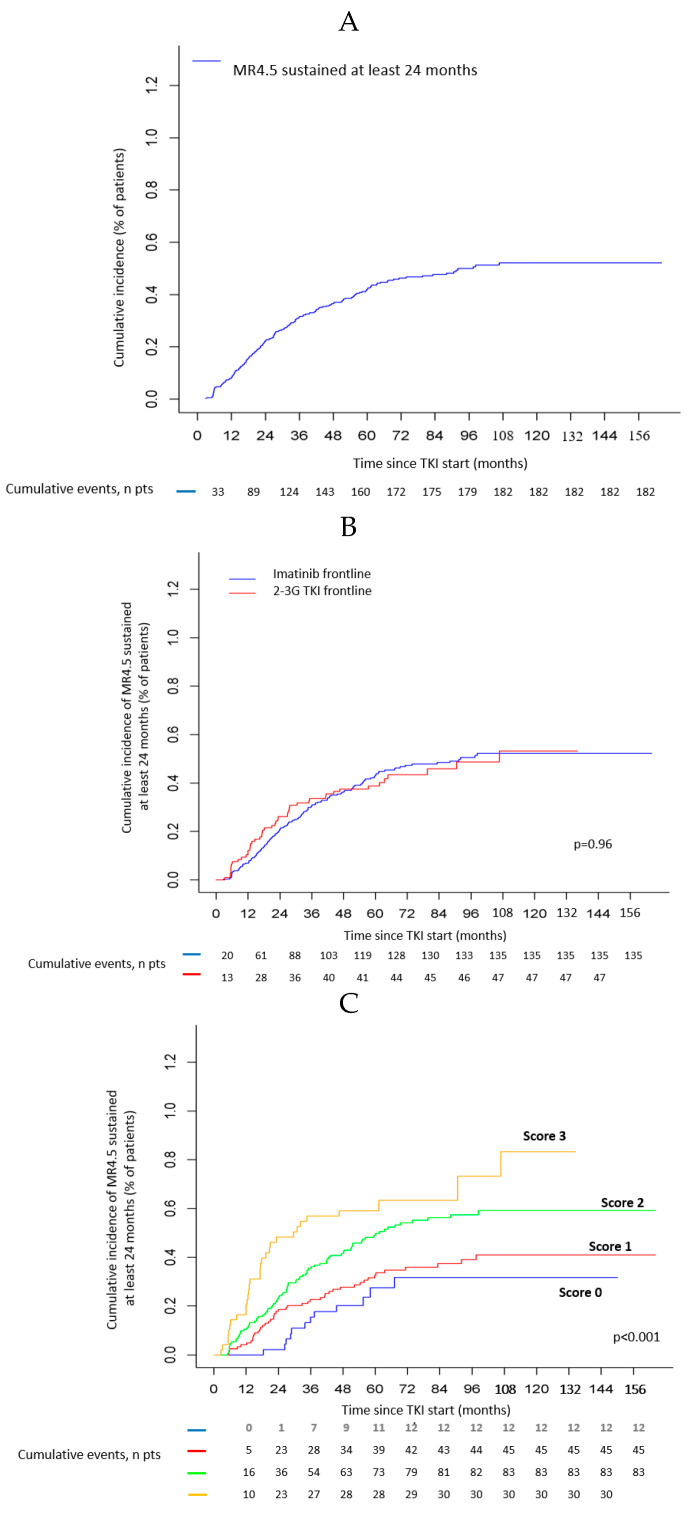
(**A**): Cumulative incidence of molecular response 4.5 log sustained for at least 24 months (N = 398 patients). (**B**): Cumulative incidence of sustained molecular response 4.5 log of at least 24 months according to first-line TKIs (Imatinib vs. 2nd or 3rd generation TKI). (**C**): Cumulative incidences of sustained MR4.5 of at least 24 months according to predictive factors. For each positive factor, female gender, B3A2 *BCR-ABL1* transcript type, and low risk ELTS score, one point is scored. The sum of points defined as the number of positive factors (0, 1, 2, or 3).

**Figure 3 cancers-12-02521-f003:**
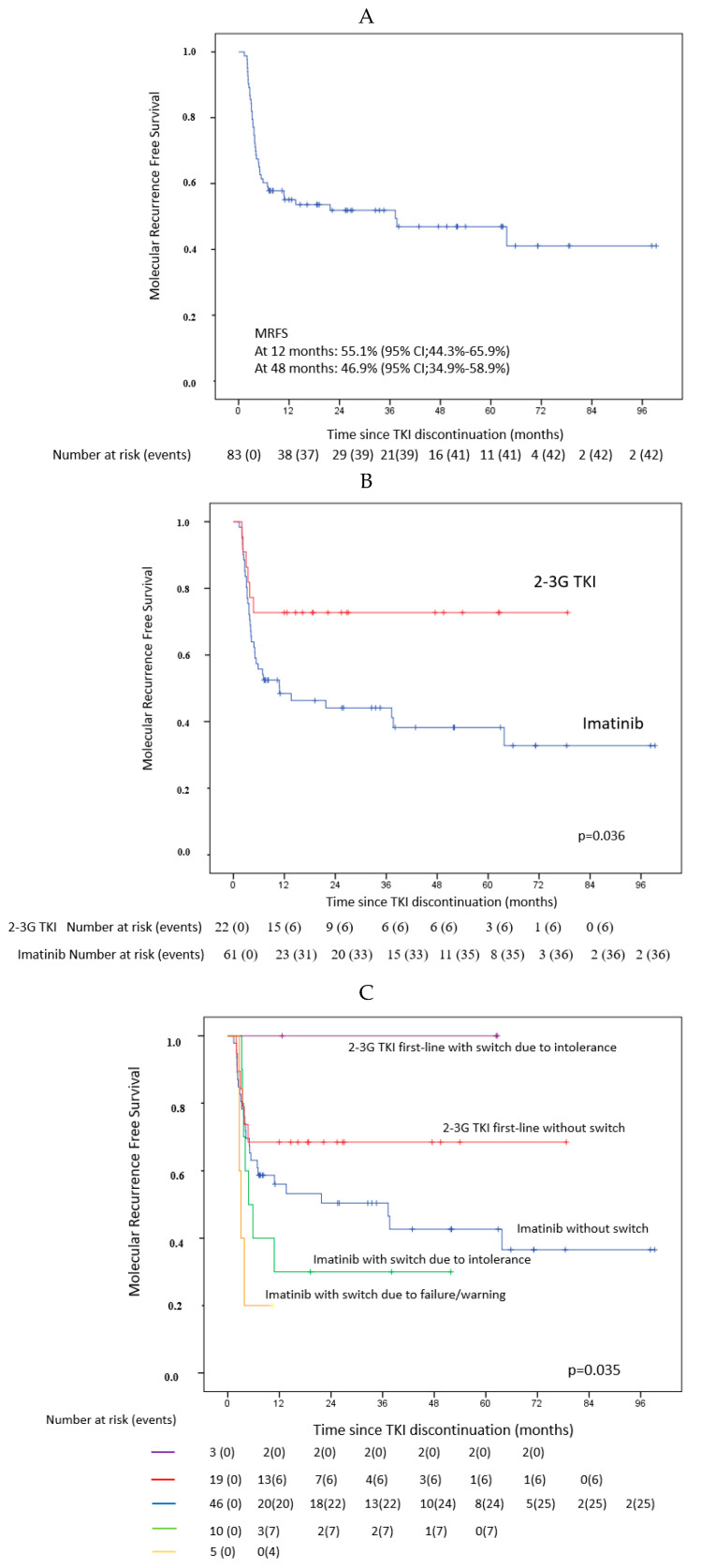
Molecular Recurrence-free survival (MRFS) (N = 83 pts). (**A**): all patients, (**B**): according to first-line TKI (Imatinib vs. 2nd or 3rd generation TKI), (**C**): according to first-line TKIs (Imatinib vs. 2nd or 3rd generation TKI) and history of switch for intolerance or insufficient efficacy (failure or warning as defined in the 2013 ELN recommendations [26]).

**Table 1 cancers-12-02521-t001:** Baseline characteristics of the study population (N = 398 patients).

Characteristics	All Patients (N = 398)	Frontline IMA Patients (N = 291)	Frontline 2–3G TKI Patients (N = 107)	*p*-Value
Median age at diagnosis, years (range)	61.6 (18.6–90.2)	63.9 (18.6–90.2)	54.0 (22.7–82.9)	<0.001
≥70, n pts (%) <45, n pts (%)	106 (26.6) 77 (19.3)	94 (32.3) 40 (13.7)	12 (11.2) 37 (34.5)	<0.001 <0.001
Gender, female, n pts (%)	168 (42.2)	123 (42.2)	45 (42.0)	0.532
Sokal risk score, n pts (%)				0.022
Low	112 (28.1)	81 (27.8)	31 (28.9)	
Intermediate	193 (49.7)	149 (51.2)	44 (41.2)	
High	76 (19.0)	46 (15.8)	30 (28.0)	
Unknown	17 (4.2)	15 (5.1)	2 (1.8)	
ELTS risk score, n pts (%)				0.054
Low	199 (50.0)	138 (47.7)	61 (56.0)	
Intermediate	122 (30.6)	98 (33.3)	24 (23.3)	
High	53 (13.3)	35 (12.0)	18 (16.8)	
Unknown	24 (6.0)	20 (6.8)	4 (3.7)	
Non-M *BCR-ABL1* transcript, n pts	3	3	0	0.373
ACA at diagnosis, n pts (%)	30 (7.5)	22 (7.5)	8 (7.4)	0.584
ACA major route at diagnosis, n pts (%)	12 (3.0)	8 (2.7)	4 (3.7)	0.410
First-line TKI type and initial daily dose, n pts (%)				
Imatinib 400	276 (69.3)	276 (94.8)	NA	
Imatinib > 400	7 (1.7)	7 (2.4)	NA	
Imatinib < 400	8 (2.0)	8 (2.7)	NA	
Nilotinib 600	69 (17.3)	NA	69 (64.4)	
Nilotinib 800	8 (2.0)	NA	8 (7.4)	
Dasatinib 100	26 (6.5)	NA	26 (24.3)	
Bosutinib 500	2 (0.5)	NA	2 (1.8)	
Ponatinib 45	2 (0.5)	NA	2 (1.8)	

Abbreviations: IMA, Imatinib; TKI, Tyrosine Kinase Inhibitor; 2–3G TKI, tyrosine kinase inhibitor of second and third generation; pts, patients; ACA, Additional Cytogenetic Abnormalities; NA, Not Applicable.

**Table 2 cancers-12-02521-t002:** Predictive factors of sustained MR4.5 for at least 24 months by univariate and multivariate Gray’s test and Fine and Gray Model Analyses.

		Cumulative Incidence of Sustained MR4.5 beyond 24 Months (*n* = 398)					
		Univariate Analysis			Multivariate Analysis		
Variable	No. Patients	SHR	95%CI	*p*	SHR	95%CI	*p*
Gender							
Male	230	-	-	-	-	-	-
Female	168	1.41	1.12–1.70	0.021	1.4	1.10–1.70	0.028
Age at diagnosis (years old)	398	0.99	0.9816–0.9984	0.24			
Type of transcript							
B2A2	130	-	-	-			
B3A2	209	1.33	1.02–1.64	0.082			
B3A2 +B2A2	23	1.11	0.46–1.76	0.75			
E8A2	1	0	−3.96	<0.001			
E19A2	2	0	−2.82	<0.001			
ACA							
No	368	-	-	-			
Yes	30	0.74	0.13–1.35	0.32			
Sokal							
Low	112	-	-	-			
Intermediate	193	0.67	0.36–0.98	0.016			
High	76	0.46	0.01–0.91	<0.001			
ELTS							
Low	199	-	-	-	-	-	-
Intermediate	122	0.46	0.089–0.83	<0.001	0.46	0.09–0.83	<0.001
High	53	0.47	−1.02	0.003	0.48	−1.01	0.004
TKI first-line							
IMA	291	-	-	-			
2–3G TKI	107	0.99	0.66–1.32	0.96			

Abbreviations: ACA, Additional Cytogenetic Abnormalities; ELTS, EUTOS Long-Term Survival; IMA, Imatinib; 2–3G TKI, 2nd and 3rd Generation Tyrosine Kinase Inhibitors.

**Table 3 cancers-12-02521-t003:** Baseline and ongoing characteristics of the treatment-free remission population (N = 83) pts).

Characteristics	All Patients (N = 83)	With MolRec (N = 42)	Without MolRec (N = 41)	*p*-Value
Median age at diagnosis, years (range)	61.8 (19.2–85.6)	61.31 (19.2–85.6)	62.69 (27.6–81.7)	0.318
Gender, female, n pts (%)	46 (55.4)	21 (50)	25 (61)	0.216
Sokal risk score, n pts (%)				0.982
Low	20 (24.1)	11 (26)	9 (22)	
Intermediate	40 (48.2)	21 (50)	19 (46)	
High	19 (22.9)	10 (24)	9 (22)	
Unknown	4 (4.8)	0 (0)	4 (9.8)	
ELTS risk score, n pts (%)				0.774
Low	46 (55.4)	26 (62)	20 (49)	
Intermediate	19 (22.9)	9 (21)	10 (24)	
High	12 (14.5)	6 (14)	6 (15)	
Unknown	6 (7.2)	1 (2)	5 (12)	
Type of Transcript				0.346
Transcript B2A2, n pts (%)	28 (33.7)	17 (41)	11 (27)	
Transcript B3A2, n pts (%)	45 (54.2)	20 (48)	25 (61)	
Transcripts B3A2 + B2A2, n pts (%)	3 (3.6)	2 (4.8)	1 (2.4)	
M *BCR-ABL1* Nos, n pts (%)	7 (8.4)	3 (7.1)	4 (9.8)	
University Hospital, n pts (%)	65 (78.3)	30 (71.4)	35 (85.4)	0.101
First-line TKI				0.010
IMA, n pts (%)	61 (73.5)	36 (85.7)	25 (60.98)	
2–3G TKI, n pts (%)	22 (26.5)	6 (14)	16 (39)	
Median time from TKI start to onset of sustained MR4.5 at least 24 months, months (range) *	21.5 (2.6–88)	21.21 (2.6–67.4)	21.85 (5.3–88)	0.339
Median time from TKI start to TKI discontinuation, years (range)	5.77 (2.8–11.2)	5.13 (2.8–10.6)	6.46 (3–11.2)	0.014
Still on first-line TKI at the date of discontinuation, n pts (%)	64 (77.1)	31 (73.8)	34 (82.9)	0.230
Sustained MR4.5 at least 24 months at the time of TKI discontinuation, n pts (%)	74 (89.2)	37 (88.1)	37 (90.2)	0.516
Median MR4.5 duration at the time of TKI discontinuation, months (range)	41.4 (13.1–113.4)	38.4 (20–104.4)	62.69 (27.6–81.7)	0.038

Abbreviations: ELTS, EUTOS Long-Term Survival; IMA, Imatinib; 2–3G TKI, 2nd or 3rd Generation Tyrosine Kinase Inhibitor; Nos, Not Otherwise Specified; War, warning; Fail, Failure according to the 2013 ELN recommendations18; MolRec, Molecular Recurrence. * Among patients with MR4.5 sustained >24 months.

**Table 4 cancers-12-02521-t004:** Predictive factors of molecular recurrence after tyrosine kinase inhibitors discontinuation by univariate and multivariate Cox regression model analyses.

		Cumulative Incidence of MolRec over Time (*n* = 83)					
		Univariate Analysis			Multivariate Analysis		
Variable	No Patients	SHR	95%CI	*p*	SHR	95%CI	*p*
Sex							
Male	37	-	-				
Female	46	0.7	0.38–1.28	0.249			
Age at diagnosis (years old)	83	0.99	0.97–1.00	0.229			
Type of transcript							
B2A2	28	-	-				
B3A2	45	0.6	0.31–1.15	0.123			
B3A2 +B2A2	3	0.88	0.20–3.83	0.866			
ACA							
No	79	-	-				
Yes	4	0.37	0.05–2.69	0.326			
Sokal							
Low	20	-	-				
Intermediate	40	0.94	0.45–1.95	0.867			
High	19	0.95	0.40–2.24	0.908			
ELTS							
Low	46	-	-				
Intermediate	19	1.07	0.50–2.29	0.867			
High	12	0.81	0.33–1.97	0.64			
First-line TKI							
IMA	61	-	-		-	-	
2–3G TKI	22	0.41	0.17–0.97	0.043	0.36	0.15–0.87	0.023
TKI switch							
No	65	-	-				
Yes	18	1.48	0.74–2.97	0.266	1.4	0.70–2.81	0.344
TKI duration (years)	83	0.87	0.75–1.01	0.066	0.85	0.73–0.98	0.029
Time to sustained MR4.5 at least 24 months (months)	78	0.99	0.98–1.01	0.467			
MR4.5 duration (months)	78	0.99	0.97–1.00	0.095			

**Table 5 cancers-12-02521-t005:** Supplemental. Predictive factors of molecular recurrence after tyrosine kinase inhibitors discontinuation by multivariate Cox regression model analyses.

		Multivariate Analysis		
Variable	No Patients	SHR	95%CI	*p*
TKI				
IMA	61	-	-	
2–3G TKI	22	0.42	0.17–1.00	0.05
Switch				
No	65	-	-	
Yes	18	0.92	0.41–2.05	0.838
MR4.5 duration (months)	78	0.99	0.97–1.00	0.095

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
