# Peer review of "Incidences of Deep Molecular Responses and Treatment-Free Remission in de Novo CP-CML Patients"

_cancers, 2020, doi:10.3390/cancers12092521_

Round 1

Reviewer 1 Report

This paper aim to study incidence and predictive factors of sustained deep molecular response in CML-CP patients de novo treated with TKI. In addition, rate and predictive factors of molecular recurrence-free survival (MRFS) were evaluated in patients who discontinued TKI.

Comments

Major comments :

  • The title is definitely confusing. Treatment free remission (TFR) and molecular recurrence free survival (MRFS) are not same endpoints. Formally, unfavourable events which have to be considered in TFR analyses are molecular recurrence and restart of treatment in remission if so. Authors should consider this important issue in both section statistical methods and results.
  • section statistical analysis:

it is stated line 138 “Variables were included in the multivariate analysis if they were significantly associated to the outcome in the univariate analysis”. What was the level of significance which was considered in these univariate analyses?

  • Please indicate how many patients were included in clinical trials before treatment discontinuation only, at treatment discontinuation only, and in both cases. Name and design of these trials should also be described (clinicaltrials.gov numbers). The point is that the monitoring of these patients could be very different according trials and may differs from monitoring outside clinical trials.

It has an impact first, with the decision of stopping treatment or not and then with the decision of restarting treatment later on. For example, according to Figure 2A, 182 pts were in sustained MR4.5 (at least 2 years). However, among them, 98 patients (53,8%) did not attempt stopping treatment. It may impact analyses exploring predictive factors of molecular recurrence. Authors please comment.

  • What do you mean by “academic” centres? You indicate that 75% of patients who stopped were in “academic centres. Does it mean that patients were referred to these centres in order to be include in TFR trials?
  • If it turns out that patients were included in different clinical trials for TFR strategies, I would not qualify your study as a “real-life” setting.
  • How many patients were less than 45 year of age? For these patients did you used the appropriate formula for the Sokal score?
  • Line 236: for consistency % in bracket should be 21% (ie 84 out of 398).

English and type writing errors to be revised:

Some examples

Line 96 : “centers” is missing

Line 134 : competing “events” (not competing “risks”)

Line 137 : “competing” events (not “competitive“ events)

Author Response

Reviewer 1

This paper aim to study incidence and predictive factors of sustained deep molecular response in CML-CP patients de novo treated with TKI. In addition, rate and predictive factors of molecular recurrence-free survival (MRFS) were evaluated in patients who discontinued TKI.

Comments

Major comments:

  • The title is definitely confusing. Treatment free remission (TFR) and molecular recurrence free survival (MRFS) are not same endpoints. Formally, unfavourable events which have to be considered in TFR analyses are molecular recurrence and restart of treatment in remission if so. Authors should consider this important issue in both section statistical methods and results.

Answer:

We do agree with this remark. We used TFR in the title of the manuscript as a generic term referring to TKI discontinuation after achieving a sustained deep molecular response. In our study, no patient restarted TKI without molecular relapse and no death was observed in patient without relapse. At the last follow-up, the 3 CML-unrelated deaths occurred after molecular relapse and TKI resumption. So TFR and MRFS do not differ. We added a sentence in the discussion and definitions section to precise this point.

  • section statistical analysis:

it is stated line 138 “Variables were included in the multivariate analysis if they were significantly associated to the outcome in the univariate analysis”. What was the level of significance which was considered in these univariate analyses?

Answer:

Variables were included in the multivariate analysis if they were significantly associated to the outcome in the univariate analysis (p<0.05) or if they were clinically relevant and based on previous published results. We added a sentence (line 138) to precise this point.

  • Please indicate how many patients were included in clinical trials before treatment discontinuation only, at treatment discontinuation only, and in both cases. Name and design of these trials should also be described (clinicaltrials.gov numbers). The point is that the monitoring of these patients could be very different according trials and may differs from monitoring outside clinical trials.

Answer:

87/398 (21%) patients were included in 11 TKI frontline clinical trials. Among tis 87 patients 6 of them have discontinued TKI inside a TKI discontinuation trial.  Among the 95 patients who stopped TKI, 24 (25%) have TKI discontinuation inside a TKI discontinuation trials. All these trials were registred in clinicaltrials.gov. Results of our study, especially molecular follow-up, were based on local molecular evaluation every 3 to 4 months and did not differed between patients treated inside or outside clinical trials.

It has an impact first, with the decision of stopping treatment or not and then with the decision of restarting treatment later on. For example, according to Figure 2A, 182 pts were in sustained MR4.5 (at least 2 years). However, among them, 98 patients (53,8%) did not attempt stopping treatment. It may impact analyses exploring predictive factors of molecular recurrence. Authors please comment.

Answer:

We do agree that all the patients who had achieved a sustained deep molecular response did not have TKI stop due to several reasons (previous history of suboptimal or failure, physician and patient decisions…). We have compared the baseline and ongoing characteristics of patients who achieved a sustained deep molecular response with and without TKI stop. Only the center type (university versus others) significantly differed between the two groups. Indeed, TKI discontinuation was first performed in discontinuation trials available only in university hospitals. Then we believe that this selection of the patients for TKI stop did not have significant impact on predictive factors of molecular recurrence. As one of a major comment of this manuscript was the high number of groups and subgroups of patients, we did not shown and discussed this point.

  • What do you mean by “academic” centres? You indicate that 75% of patients who stopped were in “academic centres. Does it mean that patients were referred to these centres in order to be include in TFR trials?

Answer: Academic was replaced by university.

  • If it turns out that patients were included in different clinical trials for TFR strategies, I would not qualify your study as a “real-life” setting.

Answer: we do agree and have suppressed this term (lines 38 and 364)

  • How many patients were less than 45 year of age? For these patients did you used the appropriate formula for the Sokal score?

Answer: 77 patients were less than 45 years old at diagnosis. We used the appropriate Sokal score. This variable was added in Table 1.

  • Line 236: for consistency % in bracket should be 21% (ie 84 out of 398).

Answer: We believed that this percentage is out of interest illustrating that all the patients who had achieved a sustained deeper molecular response do not have TKI stop. In all, 95/398 patients have stopped TKI (24% of the all cohort).

English and type writing errors to be revised:

Some examples

Line 96 : “centers” is missing

Line 134 : competing “events” (not competing “risks”)

Line 137 : “competing” events (not “competitive“ events)

Answer: the corresponding corrections have been done and several corrections  others corrections have been done.

Reviewer 2 Report

I have no significant further comment.

The manuscript improved its overall readability, although there are still some clerical errors and some misused conjunctions.

Regarding the point raised by the Authors about "CML literature" abbreviations, I am very familiar with them; nevertheless, a fluent narration requires a proper use of these abbreviations and an accurate sentence construction, rather than a pervasive american-syile acronymization.

Author Response

Several modifications have been done in the text in order to improve the manuscript

Reviewer 3 Report

In the revised paper entitled "Incidences of deep molecular responses and treatment-free remission in de novo CP-CML patients in a real-life setting", Etienne et al. gives more information to understand their results. I appreciated the statistical cohort description and the improvement in the introduction section.

I have no more comments to the text.

I have got only two doubts about results. In my opinion it is strange that frontline TKI type was not a good predictor of deep molecular response (DMR) in spite of the faster and deeper molecular responses observed with second‐generation TKIs in several studies.

On the other side, it is also strange that DMR good predictors such as BCR-ABL1 transcript type, Sokal Score and duration of TKI therapy and which are also in the ESMO recommendations for TFR, were not good predictors of Molecular Recurrence-Free Survival (MRFS).

I hope that future results in other studies will confirm these data in order to shed lights on TKI discontinuation.

Author Response

Reviewer 3

Comments and Suggestions for Authors

In the revised paper entitled "Incidences of deep molecular responses and treatment-free remission in de novo CP-CML patients in a real-life setting", Etienne et al. gives more information to understand their results. I appreciated the statistical cohort description and the improvement in the introduction section.

I have no more comments to the text.

I have got only two doubts about results. In my opinion it is strange that frontline TKI type was not a good predictor of deep molecular response (DMR) in spite of the faster and deeper molecular responses observed with second‐generation TKIs in several studies.

On the other side, it is also strange that DMR good predictors such as BCR-ABL1 transcript type, Sokal Score and duration of TKI therapy and which are also in the ESMO recommendations for TFR, were not good predictors of Molecular Recurrence-Free Survival (MRFS).

I hope that future results in other studies will confirm these data in order to shed lights on TKI discontinuation.

Answer :

We believe that the absence of any difference in terms of cumulative incidence of sustained DMR between IMA versus others TKI is mainly related to the patient populations. However, we could not excluded that switch from IMA to 2-3G TKI may contribute to this observation. Considering the risk of molecular relapse after stop, BCR-ABL1 transcript type and Sokal risk score were not always significantly associated with the risk of relapse. We confirmed the impact of TKI duration on molecular recurrence after stop and observed only a trend for MR4 .5 duration. We do agree that these results need to be confirmed on larger cohorts. 

Reviewer 4 Report

No further comments.

Author Response

Thank you very much

Round 2

Reviewer 1 Report

no more comment

This manuscript is a resubmission of an earlier submission. The following is a list of the peer review reports and author responses from that submission.

Round 1

Reviewer 1 Report

In the maniscript, the Authors investigate the possibility of TKI discontinuation in chronic myeloid leukemia, evaluating a retrospective cohort. they report 95/398 patients MR4.5-sustained eligible for treatment discontinuation, with 42 patients experiencing molecular relapse, associated with the type of therapy (imatinib vs dasatinib/nilotinib/other)

- Authors should emphasize the novelty of their work as the topic is of high interest. Also, several key references are missing.

- A striking 40% of patients rapidly progress after TKI discontinuation within 6 months, (23% after 2-3Gen TKI). Authors should expand and comment or provide a better characterization of these patients. Do they develop mutations? Being the focus of the paper, the topic is hardly investigated.

- What is the incidence of mutations in BCR-ABL? Is there a difference in the frequency/burden in those patients who switched therapy or failed to reach a molecular response?

- The manuscript interesting, but it is quite hard to read. Sentences are often complex and the use of too many acronyms makes reading quite difficult. Authors are advised to revise the whole text for a more fluent narration.

Reviewer 2 Report

In the paper entitled "Incidences of deep molecular responses and treatment-free remission in de novo CP-CML patients in a real-life setting", Etienne et al. describe disease progression in 398 chronic phase chronic myeloid leukaemia (CP-CML) patients treated with first-line TKI in order to identify predictive factors of deep molecular response (DMR) and Molecular Recurrence-Free Survival (MRFS) after TKI discontinuation. Around half of patients achieved a DMR for 2 years. They confirmed some predictors of good response such as gender, BCR-ABL1 transcript type, risk scores and type of TKI while they found only 2-3G TKI first-line as possible biomarker of treatment-free remission (TFR) but with not too strong statistical power.

The paper is well written, the results are well exposed. Unfortunately, despite of a huge number of patients, the conclusions are the confirmation of previously reports (STIM, EURO-SKI. STOP2g-TKI trials) and with no novelty.

My comments are:

  • Please make a statistical comparison between IMA frontline patients and 2-3G TKI frontline patients through Fisher or chi-square test in order to appreciate any difference between the two groups. In the conclusion you affirmed that first line TKI was a predictor of MRFS, so it is important to know all the differences between the 2 groups to interpret correctly the results.
  • Figures 2 and 3 are too blurry, and they are difficult to read. In particular the legend of figure 2C. Please, write only scores in the figure and write description in the figure caption.
  • Please make a statistical comparison between TFR population with or without MolRec through Fisher or chi-square test in order to appreciate any difference between the two groups. I think could be useful to identify characteristics of patients where TFR could be dangerous  
  • It is not clear if in TFR population there were death of disease in order to understand biomarker of bad prognosis after TFR. Please discuss it.
  • In the population several patients died for cardiovascular disease (CVD). Are there any correlations between CVD deaths and TKI treatment. Could these patients be rescued by TFR, instead of TKI side effects?

Reviewer 3 Report

The authors address an important area of interest for CML patients; prolonged treatment-free remission or TFR in de novo CP-CML patients who achieved a sustained DMR. They explored this subject in a relevant size cohort of almost 400 patients over 10 years of what the authors considered “real-life”. With a median follow-up of 7 years, 46% patients achieved a DMR sustained at least 24 months. Gender, BCR-ABL1 transcript type and Sokal and ELTS risk scores were significantly associated with a higher probability of sustained DMR. From those patients a fraction of them entered in a TFR program; interestingly, TKI first-line agent and TKI duration were the most significant factor of MRFS.

Major comments

  1. As indicated patients from clinical trials are also included. This information is lost within the numbers. Inclusion criteria’s usually select a fitter population and this kind of information is lost in this meta-analysis. This set of patients is especially enriched in the TFR (75% in academic centers (line 227), it means clinical trials?), thus the title is misleading when the authors indicates “in a real-life setting”. Additional table comparing such patients should be included and also it should be indicated more clear through the text.
  2. Statistics comparing the groups in Table 1 and Table 3 should be included. In addition, this new information from Table 1 will be the base for the comments in lines 315-18. As presented is a mere speculation, we do not know if IMA patients are significantly older than 2-3G patients.
  3. The reader will appreciate if the authors could find a way to simplify the way they make reference to the groups, subgroups, sub subgroups… In some instance is difficult to quickly understand the set of patients being described. Figure 1 is very helpful but additional simplification in the text should be addressed.

Minor comments

  1. Line 133-141 font size is different
  2. In line 171 “figure 1” should be capitalize “Figure 1”.
  3. Figures resolution is very poor and difficult to read the text.
  4. Line 219 the dot in “Table 2.” is underlined
  5. I do not understand from where the “255” patients in line 286 are coming from
  6. In line 295 the 27% corresponds to 2-3G as first line therapy why are the ones as second line therapy did not discussed?